# Determination of Organosulfides from Onion Oil

**DOI:** 10.3390/foods9070884

**Published:** 2020-07-06

**Authors:** Maranda S. Cantrell, Jared T. Seale, Sergio A. Arispe, Owen M. McDougal

**Affiliations:** 1Biomolecular Sciences Ph.D. Program, Boise State University, Boise, ID 83725, USA; marandacantrell@u.boisestate.edu; 2Department of Chemistry and Biochemistry, Boise State University, Boise, ID 83725, USA; jaredseale@u.boisestate.edu; 3Malheur County Extension Office, Oregon State University, Ontario, OR 97914, USA; sergio.arispe@oregonstate.edu

**Keywords:** yellow onion, onion oil, organosulfide, food flavoring, *Allium* cepa L.

## Abstract

Qualitative and semi-quantitative analysis of organosulfides extracted from oil obtained by steam distillation of yellow onions was performed by gas chromatography-mass spectrometry (GC-MS). The extraction efficiency of organosulfides from onion oil was evaluated across four solvents: dichloromethane; diethyl ether; n-pentane; and hexanes. Analysis of solvent extracted organosulfides by GC-MS provided qualitative results that support the use of dichloromethane over other solvents based on identification of 27 organosulfides from the dichloromethane extract as compared to 10 from diethyl ether; 19 from n-pentane; and 17 from hexanes. Semi-quantitative evaluation of organosulfides present in the dichloromethane extract was performed using diallyl disulfide as the internal reference standard. Three organosulfides were detected in the extract at ≥5 mg/kg; 18 organosulfides between 3–5 mg/kg; and six organosulfides at <3 mg/kg. The E/Z isomers of 1-propenyl propyl trisulfide were among the most prevalent components extracted from the onion oil across all solvents; and 3,6-diethyl-1,2,4,5-tetrathiane was among the most abundant organosulfides in all solvents except hexanes. The method described here for the extraction of organosulfides from steam distilled onion oil surveys common solvents to arrive at a qualitative and semi-quantitative method of analysis for agricultural products involving onions; onion oil; and secondary metabolites of *Allium* spp.

## 1. Introduction

Steam distilled onion oil is popular in the food and health industry because organosulfide components of the oil infuse succulent flavors, serve as antioxidants, suppress premature food browning, and delay the propagation of pathogenic and spoilage microorganisms [1]. One commercial method to obtain onion oil from yellow onions is by steam distillation; the onion oil is sold as a commodity to restaurants as a food flavoring ingredient, or as a therapeutic essential oil for hair follicle revitalization and homeopathic remedy for alopecia areata [2,3]. The methods for extraction of organosulfide components from steam distilled onion oil have involved the use of solid phase microextraction, headspace, or liquid–liquid extraction [4,5,6].

Solid phase microextraction involves the use of specialized fibers coated with extraction polymer or adsorptive particles embedded in a polymer that capture volatile molecules for analysis by gas chromatography-mass spectrometry (GC-MS) and has been recently used to compare the composition of the volatile organic compounds of onion and shallot [7]. The solid phase microextraction fibers are specific for volatile organic compound analysis, making selection of the correct fiber essential for optimal component analysis. Sample preparation to ensure sufficient volatile organic compound adsorbance onto the solid phase microextraction support to permit GC-MS injection concentrations suitable for analyte detection and characterization requires stringent method development. The solid phase microextraction fiber sample holder, solid phase microextraction fibers, and headspace injection system adds to the initial cost to implement this technique.

Headspace analysis of volatile organic compounds also requires the headspace assembly accessory for the GC. The volatile organic compound concentration can be a challenge as gas phase injection into the GC is inherently dilute [8,9]. Lastly, the headspace volatilization of sample, like onion oil, requires heating, which can degrade the volatile organic compounds prior to their identification [9].

Perhaps the most rudimentary and reliable process for volatile organic compound characterization from onion oil is liquid–liquid extraction. In the current study, a series of organic solvents were evaluated for qualitative and quantitative extraction of organosulfides from onion oil obtained by steam distillation of yellow onions (*Allium* cepa L.), based on the method of Tocmo et al. [10]. The four organic solvents assessed for liquid–liquid extraction of organosulfides from onion oil were dichloromethane (DCM), diethyl ether (DEE), n-pentane, and hexanes. Selection of these solvents was based on literature precedent in studies of the anticancer and antibacterial properties associated with the organosulfide constituents of garlic (*Allium* sativum L.) and onions (*Allium* cepa L.) [11,12,13,14].

The solvents most commonly reported for organosulfide extraction from *Allium* oils are DCM and DEE [10,15,16,17]. n-Pentane and hexanes are less commonly reported solvents for extraction of organosulfides, but there are studies that have demonstrated their effectiveness with garlic and other *Allium* spp [18,19]. The organosulfide profile of onions has been widely studied, along with the effects on organosulfide formation upon treatment of onions to conditions including boiling, freezing, freeze drying, and even different methods of homogenization in various solutions [4,12,20,21,22]. However, the extraction solvent used for onion oil from yellow onion has been little studied. Here, we report a comparison of four commonly used organic solvents to extract organosulfides in onion oil derived from steam distilled yellow onions to determine which solvent provides superior qualitative and semi-quantitative results upon analysis by GC-MS. While qualitative identification of organosulfide constituents in onion oil has been performed, quantitative yields correlated to extraction solvent have not been reported [4,6]. An article describing the optimization of GC methods for compound identification of volatile organic compounds extracted from raw onion has also been published, but the study did not address quantitative analysis on steam distilled onion oil [19]. In the present study, we sought to determine the optimal extraction solvent for steam distilled onion oil to provide the best qualitative and semi-quantitative characterization of organosulfides present in the oil. The term semi-quantitative is used to refer to organosulfide quantitation using a single calibration standard, diallyl disulfide, to approximate the concentration of all organosulfide constituents in each extract. It is envisioned that food processors seeking to implement quality control measures on onion oil food flavoring products may adopt the method presented here.

## 2. Materials and Methods

### 2.1. Chemicals and Materials

Dimethylchloride (DCM; HPLC grade > 99%), diethyl ether (DEE; anhydrous, HPLC grade > 99%), n-pentane (anhydrous, >99%), hexanes (>98.5%), and diallyl disulfide (DADS, >80%) were purchased from Fisher Scientific (Hampton, NH, U.S.A.). Polytetrafluoroethylene (PFTE; 0.45 μm) filters were also purchased from Fischer Scientific.

### 2.2. Sample Preparation

Yellow onions of species *Allium* cepa L. were purchased from a local supermarket and kept at room temperature until use. The onions were prepared by pealing their skins, removing the ends, and rinsing with 18 megaohm (MΩ) nanopure water. Approximately 155 g of freshly peeled onion was homogenized using a Hamilton Beach 10-cup Food Processor Model 70,730 with enough nanopure water to cover the onions. The homogenate was placed in a 500 mL round bottom flask and steam distilled for 3.5 h. The distillation process was repeated with fresh onion until approximately 500 mL of steam distilled onion oil was obtained. From this 500 mL stock of onion oil, approximately 35 mL of the milky distillate was transferred to a separatory funnel, and combined with equivalent amounts of extraction solvent, either DCM, DEE, n-pentane, or hexanes. The separatory funnel was cocked and shaken with intermittent pressure release until the buildup of gas no longer occurred. The organic layer was collected, and the milky aqueous layer underwent two additional extractions of 35 mL solvent each, until the aqueous layer appeared clear. The organic extracts were combined, and the solvent evaporated overnight in a chemical fume hood, leaving the flask with the desired yellow–green onion oil residue. The oil was weighed and dissolved in 1 mL DCM, then filtered through a pre-wetted 0.45 μm PFTE membrane, spiked with 2 μL of a 1% (*v*/*v*) solution of DADS in DCM as an internal standard, and analyzed by GC-MS.

### 2.3. GC-MS Method & Compound Identification

The GC method, adapted from Tocmo et al., was performed using a Thermofisher Trace GC Ultra system equipped with a fused silica TG-5MS (30 m × 0.25 mm i.d., 0.25 µm) column [7]. The injection port temperature was 250 °C, and the run method began with an initial temperature held at 50 °C for 3 min, raised at 3 °C/min to 150 °C, held at 150 °C for 3 min, and finally ramped at 25 °C/min to a final temperature of 250 °C, where the temperature was held for 5 min. Helium was the carrier gas with a flow rate of 1 mL/min. One µL of sample was injected at a split ratio of 1:10. The GC system was interfaced with a Thermofisher ITQ900 Quadrupole Ion Trap Mass Spectrometer with an ion source temperature of 200 °C and a mass range of 40–500. Quantification was performed by comparison to a standard curve generated from analysis of five known concentrations of DADS standards in DCM. For initial molecule identification, the National Institute of Standards and Technology (NIST) mass spectral library version 2.2 was used [23]. Chromatographic peaks interpreted as molecules with low probability scores (<10) and/or no consensus identity among triplicate samples were omitted, as well as those molecules suspected to be by-products from decomposition due to the GC injection port temperature of 250 °C used to volatilize the sample.

### 2.4. Statistical Analysis

Results were expressed as means ± standard deviations (SD) of three separate extractions per solvent used. Poisson regression was carried out to evaluate the effect of solvent on the amount of organosulfides detected. From the regression analysis, a probability value (*p*-value) was calculated, where a statistically significance result correlates to a *p*-value ≤ 0.05. Differences between the yield of organosulfides in the different solvents were analyzed by an ANOVA (*p* ≤ 0.05) and post-hoc analysis using Tukey’s test to determine significant differences among the means (*p* ≤ 0.05). Statistical analyses were carried out using IBM SPSS Version 25.0 (Armonk, NY, USA).

## 3. Results and Discussion

### 3.1. Organosulfide Determination and Quantitation in Steam Distilled Onion Oil

Figure 1a–d shows chromatograms for the organosulfides extracted with DCM, DEE, n-pentane, and hexanes, respectively. The numbers over the peaks in each chromatogram in Figure 1, correspond to the organosulfide constituents present in the extract. Table 1 shows organosulfides identified by comparison to molecules presented in literature references together with their GC retention time, molecular formula, compound identity, and characteristic *m*/*z* fragments. Eleven of the 29 tabulated organosulfides represent pairs of stereoisomers, with only one pair, (*E*/*Z*)-1-propenyl propyl trisulfide being definitively differentiated from one another prior [6]. The statistical significance for the amount of organosulfide present was determined by a one-way ANOVA and Poisson regression (SPSS, v. 25.0) where *p* < 0.05 (Table 2). The DADS internal standard is commercially available at a purity of 78%, where the other 22% is accounted for by diallyl sulfide and diallyl trisulfide. Calibration of the DADS standard by GC-MS at five different concentrations permitted qualitative and quantitative analysis of DADS, diallyl sulfide, and diallyl trisulfide. The diallyl sulfide and diallyl trisulfide were not utilized for the semi-quantitative assessment of organosulfides in onion oil, and were not included in Table 1. Figure 2 shows the total amount of organosulfides extracted by each solvent in total (Figure 2a), and the composition of the oil extract segregated by the amount of cyclic, acyclic di-, acyclic tri-, and polysulfides, respectively (Figure 2b). The amount of cyclic organosulfides present in the oil extract across all of the organic solvents was consistently the abundant species, followed by acyclic trisulfides, acyclic disulfides, and polysulfides, respectively. For the purposes of interpreting these data, polysulfides were characterized as molecules containing four or more sulfur atoms.

### 3.2. Solvent Extraction Efficiency

Solvent extraction was carried out on approximately 35 mL of steam distilled onion oil in a separatory funnel with either DCM, DEE, n-pentane, or hexanes; oil extraction was performed in triplicate to provide the standard deviations presented in Table 2. The concentration of organosulfides extracted from each 35 mL onion oil sample were deemed statistically relevant across triplicate measurement provided *p* < 0.05 (see Table 2). The type of organosulfide extracted from *Allium* oils has been reported to vary depending on processing conditions [4,10]. For the present study, onions were prepared as described in the methods; steam distillation of several batches of yellow onions provided a combined oil volume of approximately 500 mL of stock solution. DCM extraction provided the most unique organosulfide constituents at **27** and the highest abundance of components at 109 mg/kg, followed by hexanes (20 organosulfides at a concentration of 68 mg/kg), n-pentane (19 organosulfides at a concentration of 78 mg/kg), and lastly DEE (10 organosulfides at a concentration of 38 mg/kg; Figure 1). Of the four solvents evaluated for qualitative and semi-quantitative determination of organosulfide content from onion oil, DCM was concluded to be the best. While semi-quantitative data are reasonably consistent across solvent types, qualitative data are better for DCM (Table 2). Statistical significance determined by Tukey’s Test indicates that DEE extraction provided the best significance when compared to the other solvents (Figure 2). The organosulfide extract obtained with DEE as the solvent was found to have significance (*p* ≤ 0.05) to the other solvents. Additionally, organosulfide extract obtained with DCM was found to have significance (*p* ≤ 0.05) to pentane. However, no other solvent was found to be significantly similar to any other solvent (i.e., hexanes with n-pentane, or hexanes with DCM, etc.) In summary, the use of DCM afforded the extraction of a greater variety and quantity of organosulfides from the onion oil, while DEE provided the greatest statistical significance when compared to other solvents. 

#### 3.2.1. Dichloromethane Extraction

Twenty-seven organosulfides were identified from the DCM extract, making this solvent the best for qualitative assessment of onion oil components (Figure 1a). The amount of organosulfides obtained by DCM extraction was determined to be an average of 108.72 ± 0.72 mg/kg (Figure 2a). The three organosulfides present in the DCM extract at high concentration of ≥5 mg/kg were 1-propenyl propyl trisulfide (E and Z isomer), and 3,6-diethyl-1,2,4,5-tetrathiane (compounds **18**, **19**, **29**). These three organosulfides have been evaluated by the Flavor and Extract Manufacturers Association (FEMA) and have been found to be generally regarded as safe (GRAS) for use as food flavoring agents [2,35]. The high abundance of these three organosulfides is to be expected from onion oil, as they are the main contributors to the desirable food taste and aroma attributes of the oil. A total of 19 organosulfides were present at medium concentrations of 3–5 mg/kg, and the remaining five organosulfides were present at concentrations less than 3 mg/kg. While consumption of organosulfides from onions can be beneficial in moderation, some of the components, like dipropyl disulfide (compound **5**) and (*E*/*Z*)-1-propenyl propyl trisulfides (compounds **18** and **19**), have been identified as the causative agent of hemolytic anemia when ingested at high amounts in rats and dogs [36,37]. These particular organosulfides are responsible for the recommendation that cattle not receive more than 25% dry weight feed of onions [38]. The use of DCM for liquid–liquid extraction, while superior to other solvents, is not without limitation due to the hazard associated with DCM inhalation exposure correlating to high incidence of tumor formation in rats and mice [39]. In humans, DCM is absorbed through the skin and inhalation leads to increased concentrations in the blood and adipose tissue [39]. Therefore, caution must be taken when working with this solvent.

#### 3.2.2. Diethyl Ether Extraction

Ten organosulfides were identified from the DEE extract by GC-MS analysis (Figure 1b), resulting in a total amount of material at 38.06 ± 1.87 mg organosulfide per kg onion (Figure 2a). Of these constituents, no organosulfides were present at >5 mg/kg, nine were present at 3–5 mg/kg, and only one was present at <3 mg/kg. The same three most abundant organosulfides from the DCM extraction were among the most prevalent in the DEE extracts ((*E*/*Z*)-1-propenyl propyl trisulfide and 3,6-diethyl-1,2,4,5-tetrathiane). Interestingly, DEE pulled out two organosulfides that DCM did not. These components were 5-methyltetrathiane and hexathiane. 5-Methyltetrathiane can be used as a food flavoring agent, but it is not recommended for use as a fragrance due to distasteful odor, and it is a product of the thermal degradation of organosulfides [40]. Hexathiane, also known as elemental sulfur or hexasulfur, has previously been identified in onions [41]. DEE has been widely used for solvent extraction of *Allium* spp. oils and fresh vegetables; however, the literature appears to shy away from DEE use for onion oil extraction specifically [10,16,42,43]. Based on our assessment of DEE, it is not an effective solvent for the extraction of organosulfides from steam distilled onion oil. The DEE extract provided only ten organosulfides from onion oil. DEE is a solvent commonly used in mouse studies for anesthesia; DEE is known to cause light-headedness and it is highly flammable [44]. As with DCM, caution must be used when working with DEE.

#### 3.2.3. n-Pentane Extraction

A total of 19 compounds were identified from the n-pentane extract with a yield of components amounting to 78.34 ± 3.71 mg/kg (Figure 2a). No organosulfides were present at above 5 mg/kg, 17 were present at 3–5 mg/kg, and only two were present at <3 mg/kg. Similar to DCM and DEE, the presence of *E* and *Z* isomers of 1-propenyl propyl trisulfide and 3,6-diethyl-1,2,4,5-tetrathiane was among the most abundant components (compounds **18**, **19**, **29**). Commensurate with DEE extraction and absent from the DCM extraction, both 5-methyltetrathiane and hexathiane were present at moderate levels of between three and five mg/kg (compounds **21** and **25**). (*E*)-1-Propenyl methyl disulfide and 3,6-diethyl-1,2,4,5-tetrathiane were determined to be only present in DCM and n-pentane (compound **3** and **28**). Both (*E*)-1-propenyl methyl disulfide and 3,6-diethyl-1,2,4,5-tetrathiane are recommended for use as flavoring agents, but not for fragrance [43]. n-Pentane had nearly double the propensity to extract different organosulfides as DEE but had significantly lower quantitative recovery of constituents as compared to DCM.

#### 3.2.4. Hexanes Extraction

Twenty organosulfides were identified in the hexanes extract with a total concentration of 67.98 ± 4.64 mg/kg (Figure 2a). No organosulfides were found to be present at greater than 5 mg/kg, 15 were present at 3–5 mg/kg, and three were present at <3 mg/kg. Similar to the other three solvents, the *E* and *Z* isomers of 1-propenyl propyl trisulfide was detected to be a major component of the extract (compounds 18 and 19). Present in the other three solvents, but absent from the hexanes extract were 3,4-dimethylthiophene and 3,6-diethyl-1,2,4,5-tetrathiane (compounds **1** and **29**). Hexanes and n-pentane were comparable to each other in organosulfide extraction efficiency based on number and amount of organosulfides, but neither was comparable to DCM.

### 3.3. Solvent Extraction

The extraction solvents evaluated in this study were consistent with those reported for extraction of essential oils obtained from *Allium* genus specimens [7,10,12,45]. Colina-Coca et al. used headspace to determine organosulfide content of diced, freeze-dried, and pulverized onion. They used semi-quantitation by dividing the compound abundance by the internal standard (allyl methyl sulfide) abundance rather than generation of a standard curve [46]. Boelens et al. used headspace analysis to detect 46 unique organosulfides from onion oil [6]. However, quantitative analysis by headspace was not performed, nor mass spectrum fragmentation patterns for the organosulfides obtained. Some recent publications have quantified certain organosulfide from *Allium*. In one example, DEE extraction of steam distilled shallot oil reported a concentration of dipropyl disulfide of 2.27 ± 0.18 mg/kg, which, in comparison to our 3.32 ± 0.17 mg/kg, was reasonably comparable given that onions and shallots are likely to vary in their organosulfide content [10]. Another study reported the quantification of (*E*)-1-(methyldisulfanyl)prop-1-ene in onion powder to be 3.89 ± 0.04 mg/kg, which in comparison to our 3.98 ± 0.45 mg/kg for compound 3, is a percent deviation of 2.3% [4]. In comparison, we have obtained semi-quantitative results for 29 organosulfide components in onion oil (Table 2) with statistical significance using ANOVA (*p* < 0.05) for four of the compounds (compounds **1**, **18**, **19** and **21**). Our results correlate well with previously published works in both qualitative and semi-quantitative measures. 

### 3.4. Organosulfide Profile of Steam Distilled Onion Oil

Organosulfides are secondary metabolites generated from the thermal degradation of S-alkenyl cysteine sulfoxides (ACSO’s) and are a type of phytonutrient shown to be beneficial in health research [20,34,35,39,40]. Of the 29 identified organosulfides, the most abundant were cyclics and/or di- and trisulfides such as 4-methyl-1,2,3-trithiolane (4MT), (*E*/*Z*)-1-propenyl propyl trisulfide (PPT), 2,3-dimethyl-5,6-dithiabicyclo[2.1.1]hexane 5,5-dioxide (DDHD), and 3,6-diethyl-1,2,4,5-tetrathiane (3,6-DT; Table 2). (*E*/*Z*)-1-Propenyl propyl trisulfide, compounds 18 and 19, were the most abundant organosulfides identified consistently across the four solvents evaluated. These compounds are responsible for the aroma of onion oil, and their abundance largely depends on the harvesting conditions in which they are grown [26].

The main advantage to solvent extraction of organosulfides versus either SPME or HS methods is that solvent extraction sample preparation is easy, efficient and provides semi-quantitative results. We have shown that estimates of organosulfide quantitation can be achieved with statistically significant results, as presented in Table 2. From the survey of four solvents, it was found that only seven of 29 total organosulfides identified, were detected in every solvent (compounds **4**–**6**, **8**, **10**, **18**, **19**). Even with the superior performance of DCM to extract 27 organosulfides from onion oil, this solvent still failed to detect two of the components identified from the other three solvents, compounds 21 and 25. It is the intent of this study to provide the benefits and limitations of solvent extraction of organosulfides from onion oil to inform quality assurance measures for food flavoring products.

## Figures and Tables

**Figure 1 foods-09-00884-f001:**
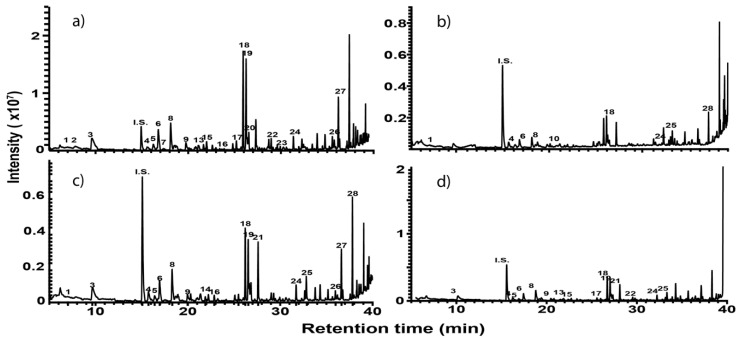
Chromatograms of organosulfides extracted from steam distilled onion oil using (**a**) dichloromethand (DCM), (**b**) diethylether (DEE), (**c**) n-pentane, and (**d**) hexanes. Chromatogram peak numbers correspond to sample number in Table 1 and Table 2.

**Figure 2 foods-09-00884-f002:**
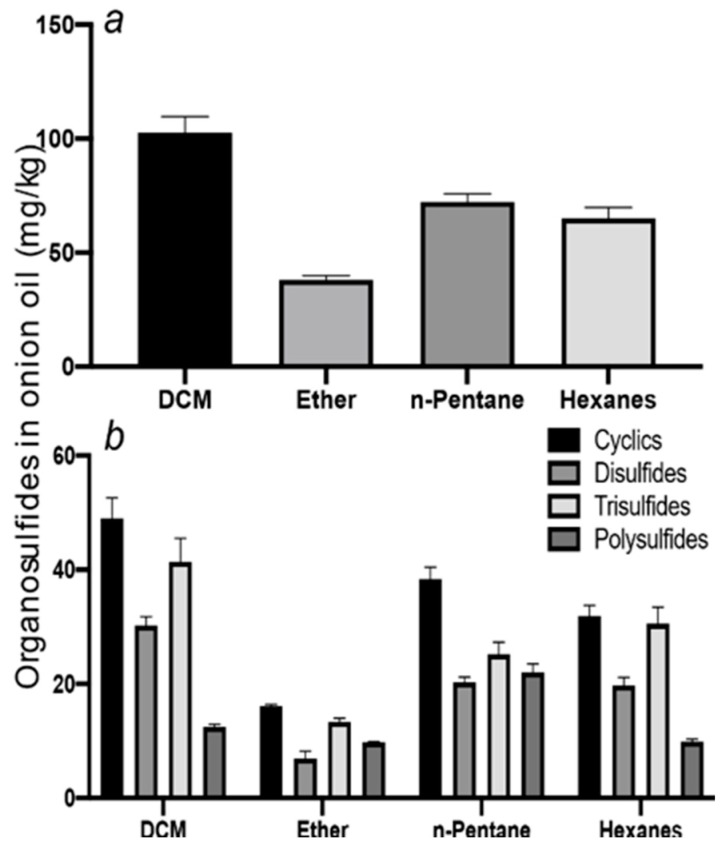
(**a**) Total amount of organosulfide obtained by solvent extraction, and (**b**) organosulfide structure type as a function of solvent used for extraction. Data are expressed as means ± SD. The difference among solvents in (**a**) were analyzed by Tukey’s test with a significance of *p* ≤ 0.05.

**Table 1 foods-09-00884-t001:** Predicted organosulfide structures and probability scores.

No ^a^	r.t. (min)	Compound Name, Formula	Structure	Probability Score ^b^	*m*/*z* (%) ^c^	Identification ^d^
**1**	7.02	3,4-Dimethylthiophene, C_6_H_8_S	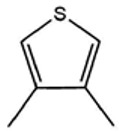	10.76	111(100),112(66),97 (56),50(29), 77(27),59(26), 106(25),113(11)	MS, [4,6,24,25]
**2**	7.87	2,4-Dimethyl Thiophene, C_6_H_8_S	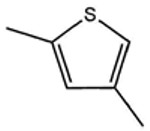	21.13	111(100),112(61),97 (41),74(39), 116(34),50(21), 77(21),113(9)	MS, [4,6]
**3**	9.65	(*E*)-1-(Methyldisulfanyl)prop-1-ene, C_6_H_8_S_2_	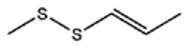	32.84	72(100),120(74), 72(42),103(27),91 (22),71(21), 104(12),121(4)	MS, [6,26]
**4**	15.73	(*E*)-Propenyl Propyl Disulfide, C_6_H_12_S_2_	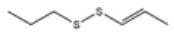	46.78	106(100),148(52),63 (21),73(20), 72(18),77(16), 59(15),149(5)	MS, [4,6,27]
**5**	16.40	Dipropyl Disulfide, C_6_H_14_S_2_	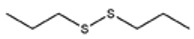	52.04	150(100),108(54),65 (17),74(15), 148(14),117(10), 75(10),151(8)	MS, [4,6,28]
**6**	16.95	Allyl Isobutyl Disulfide, C_7_H_12_S_2_	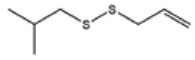	49.47	73(100),105(42), 148(39),83(15), 71(9),72(6),74(5), 149(3)	MS, [27]
**7**	17.55	2-Ethyl-1,3-Dithiane, C_6_H_12_S_2_	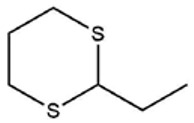	32.08	119(100),85(22), 148(21),58(12), 73(10),121(9), 75(6)	MS, [29,30]
**8**	18.31	4-Methyl-1,2,3-Trithiolane, C_3_H_3_S_6_	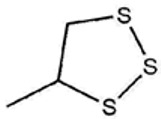	86.36	137(100),73(55), 63(14),139(14), 59(13),154(12), 74(12),95(9) *	MS, [12,13]
**9**	19.97	3,4-Dimethyl-2,3-Dihydro-2-Thiophenethiol, C_6_H_10_S_2_	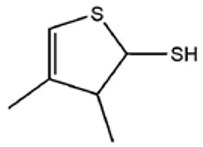	80.51	144(100),111(77),117(49),146(28), 143(18),77(16), 99(16),129(12)	MS
**10**	20.30	4H-1,2,3-Trithiin, C_3_H_4_S_3_	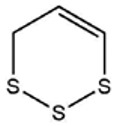	14.51	70(100),135(68), 103(59),72(15), 143(14),63(13), 144(12),137(4)	MS, [14,31]
**11**	20.53	2-Ethylidene-1,3-Dithiane, C_6_H_10_S_2_	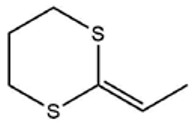	59.44	146(100),71(33), 104(33),103(31), 75(22),113(20), 85(14),147(10)	MS, [15,32]
**12**	21.03	1-(Methylthiopropyl) Methyl Disulfide, C_5_H_12_S_3_	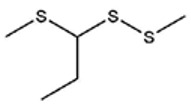	24.54	89(100),61(53), 73(19),88(17), 146(14),71(13), 103(13),161(10) *	MS, [5,16,24,25]
**13**	21.33	Methyl-*trans*-1-propenyl trisulfide	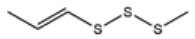	27.50	146(100),152(80),103(77),87(64), 117(58),71(43), 85(37),154(11)	MS, [6,16,28,33,34]
**14**	21.89	3-Ethyl-5-methyl-1,2,4-trithiolane, C_4_H_8_S_3_ (isomer of 15)	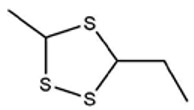	13.50	166(100),101(68),59 (54),63(25), 69(22),168(14),137 (13),102(13)	MS, [17]
**15**	22.22	3-Ethyl-5-methyl-1,2,4-trithiolane, C_5_H_10_S_3_ (isomer of 14)	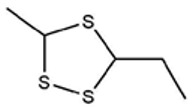	92.55	166(100),101(63),58 (51),63(24), 69(21),168(14), 102(13),73(12)	MS
**16**	23.66	2-Methyl-4-Methylsulfanyl-2,3-Dihydrothiophene, C_6_H_10_S_2_	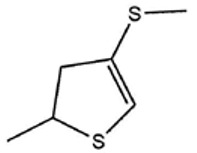	26.86	146(100),131(74),71 (41),73(24), 77(22),59(22), 55(21),79(19)	MS
**17**	25.48	5-Methyl tetrathiane, C_3_H_6_S_4_ (isomer of 21)	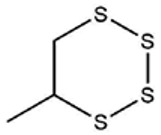	94.32	169(100),106(78),63 (36),171(19), 59(14),127(14), 72(12),170(6)	MS, [10,29]
**18**	26.22	(*E*)-1-Propenyl Propyl Trisulfide, C_6_H_12_S_3_ (isomer of 19)	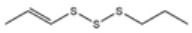	65.27	180(100),182(73),115(70),75(26), 106(18),58(18), 55(17),181(9)	MS, [6,24]
**19**	26.55	(*Z*)-1-Propenyl Propyl Trisulfide, C_6_H_12_S_3_ (isomer of 18)	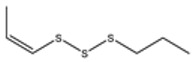	81.35	180(100),115(68),81 (19),74(17), 58(16),106(16), 182(13),181(9)	MS, [6,24]
**20**	26.84	3,5-Diethyl-1,2,4-Trithiolane, C_6_H_12_S_3_	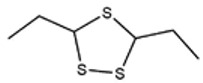	30.68	116(100),180(38),74 (32),73(22), 114(12),118(11), 182(6),181(3)	MS, [10]
**21**	27.61	5-Methyltetrathiane, C_3_H_6_S_4_ (isomer of 17)	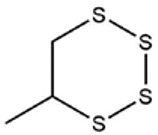	95.88	169(100),106(49),127(26),63(20), 171(18),59(9), 73(7),170(6)	MS, [10,25,33]
**22**	29.28	1-(Methylthio)Propyl Propyl Disulfide, C_7_H_16_S_3_ (isomer of 23)	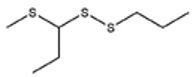	31.78	89(100),61(47),73 (35),194(14), 55(10),70(8), 88(7),59(6)	MS, [16,18,24,25]
**23**	30.28	1-(Methylthio)Propyl Propyl Disulfide (isomer of 22), C_7_H_16_S_3_	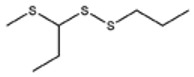	38.72	89(100),61(47), 73(35),194(14), 55(10),70(8), 88(7),59(6)	MS, [16,18,24,25]
**24**	31.72	2,3-Dimethyl-5,6-Dithiabicyclo [2.1.1]hexane 5,5-Dioxide, C_6_H_10_O_2_S_2_	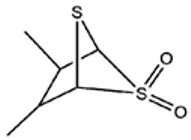	86.57	113(100),99(70), 177(34),79(23), 77(22),111(21), 97(16),65(16)	MS, [15,24]
**25**	32.88	Hexathiane, S_6_	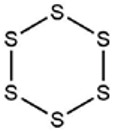	74.50	191(100),63(29), 193(28),127(26), 130(4),192(4), 96(4),65(3)	MS, [19,23]
**26**	35.99	Dipropyl Tetrasulfide, C_6_H_14_S_4_	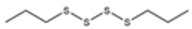	82.22	214(100),73(34), 108(27),216(17), 75(15),117(14), 171(13),69(11)	MS, [18,25,28]
**27**	36.64	3,6-Diethyl-1,2,4,5-Tetrathiane, C_6_H_12_S_4_ (isomer of 28, 29)	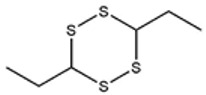	35.67	73(100),115(91), 147(65),211(29), 74(17),138(14), 81(12),59(9)	MS, [23]
**28**	37.84	3,6-Diethyl-1,2,4,5-Tetrathiane, C_6_H_12_S_4_ (isomer of 27, 29)	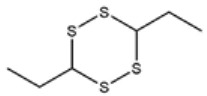	35.28	73(100),115(94), 147(60),211(22), 74(19),81(14), 137(13),180(10)	MS
**29**	37.84	3,6-Diethyl-1,2,4,5-Tetrathiane, C_6_H_12_S_4_ (isomer of 27, 28)	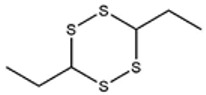	35.28	73(100),115(94), 147(60),211(22), 74(19),81(14), 137(13),180(10)	MS

^a^ Refers to peaks in Figure 1. MS ^b^ Determined using the National Institute of Standards and Technologies (NIST) mass spectral library version 2.2. ^c^ Determined with a ThermoFisher ITQ900 quadrupole ion trap mass spectrometer (Thermo, CA, USA), TG-5MS (30 m × 0.25 mm i.d., 0.25 μm) column, *m*/*z* with relative intensity in parenthesis in decreasing order. ^d^ Initial identification using MS with reference precedence * Molecular ion peak not found.

**Table 2 foods-09-00884-t002:** Semi-quantitative organosulfide content from each extraction solvent.

	Organosulfide Content (mg/kg) in Each Solvent			
No *^α^*	DCM	DEE	*n*-Pentane	Hexanes
**1**	3.03 ± 0.06 ^a^	2.94 ± 0.04 ^b^	2.90 ± 0.01 ^a^	n.d.
**2**	3.36 ± 0.17 ^a^	n.d.	n.d.	n.d.
**3**	3.98 ± 0.45 ^a^	n.d.	3.70 ± 0.29 ^b^	n.d.
**I.S.**	5.29 ± 1.09 ^a^	5.08 ± 0.26 ^b^	5.13 ± 0.56 ^c^	4.96 ± 0.08 ^d^
**4**	3.27 ± 0.02 ^a^	3.71 ± 1.1 ^b^	3.22 ± 0.03 ^c^	3.16 ± 0.09 ^d^
**5**	3.32 ± 0.16 ^a^	n.d.	3.14 ± 0.09 ^b^	3.10 ± 0.13 ^c^
**6**	4.26 ± 0.62 ^a^	3.22 ± 0.16 ^b^	3.82 ± 0.31 ^c^	3.64 ± 0.49 ^d^
**7**	2.99 ± 0.05 ^a^	n.d.	n.d.	n.d.
**8**	4.30 ± 0.56 ^a^	3.21 ± 0.15 ^b^	3.93 ± 0.3 ^c^	3.55 ± 0.68 ^d^
**9**	3.40 ± 0.21 ^a^	n.d.	3.21 ± 0.11 ^b^	3.14 ± 0.17 ^c^
**10**	2.99 ± 0.02 ^a^	3.14 ± 0.31 ^b^	2.98 ± 0.02 ^c^	2.96 ± 0.01 ^d^
**11**	2.93 ± 0.01 ^a^	n.d.	n.d.	n.d.
**12**	3.16 ± 0.14 ^a^	n.d.	n.d.	n.d.
**13**	3.30 ± 0.08 ^a^	n.d.	n.d.	3.07 ± 0.15 ^b^
**14**	3.08 ± 0.09 ^a^	n.d.	3.00 ± 0.06 ^b^	2.96 ± 0.06 ^c^
**15**	3.23 ± 0.19 ^a^	n.d.	3.10 ± 0.11 ^b^	3.02 ± 0.09 ^c^
**16**	2.90 ± 0.02 ^a^	n.d.	n.d.	n.d.
**17**	3.06 ± 0.23 ^a^	n.d.	3.03 ± 0.07 ^b^	2.98 ± 0.06 ^c^
**18**	5.66 ± 1.35 ^a^	3.41 ± 0.22 ^b^	4.06 ± 1.09 ^b^	4.27 ± 0.84 ^c^
**19**	5.84 ± 1.30 ^a^	3.56 ± 0.30 ^a^	4.84 ± 0.74 ^b^	4.40 ± 0.76 ^c^
**20**	3.61 ± 0.30 ^a^	n.d.	n.d.	3.26 ± 0.26
**21**	n.d.	3.17 ± 0.28 ^a^	3.86 ± 0.14 ^b^	3.67 ± 0.32 ^a^
**22**	3.23 ± 0.09 ^a^	n.d.	n.d.	3.02 ± 0.13 ^b^
**23**	2.97 ± 0.05 ^a^	n.d.	n.d.	n.d.
**24**	3.20 ± 0.11 ^a^	n.d.	3.20 ± 0.10 ^b^	3.15 ± 0.13 ^c^
**25**	n.d.	3.21 ± 0.07 ^a^	3.28 ± 0.11 ^b^	3.24 ± 0.08 ^c^
**26**	3.19 ± 0.16 ^a^	n.d.	3.07 ± 0.07 ^b^	n.d.
**27**	4.50 ± 0.68 ^a^	n.d.	n.d.	n.d.
**28**	3.20 ± 0.11 ^a^	n.d.	4.20 ± 0.45 ^b^	n.d.
**29**	5.37 ± 1.14 ^a^	3.41 ± 0.26 ^a^	4.60 ± 0.68 ^b^	n.d.

^α^ Corresponds to compound numbers in Figure 1. Concentrations and standard deviations expressed as means ± SD. The difference among solvents were analyzed by Tukey’s test with a significance of *p* ≤ 0.05. Values across rows without a common letter (a–d) are significantly different. n.d.: not detected. DCM = dichloromethane; DEE = diethyl ether; SD = standard deviation.

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
