# Peer review of "Determination of Organosulfides from Onion Oil"

_foods, 2020, doi:10.3390/foods9070884_

Round 1
Reviewer 1 Report
Cantrell and coworkers report a qualitative and semi-quantitative analysis of organosulfides extracted from steam distilled yellow onion oil through gas chromatography-mass spectrometry. Even though the extraction of organosulfides is performed with standard liquid-liquid extraction, the obtained results are noteworthy and of interest to journal readers. The paper is therefore suitable for publication in Foods after minor revision. I have however a few comments and it would be worth to take them into account.
- Most of the manuscript is written in a style that more closely resembles that of a scientific report rather than a paper. I, therefore, recommend improving the style.
- Abstract section (rows 17-18): “the result was three organosulfides present…”, I suggest rephrasing this sentence in a more accurate form.
- Row 115: it is worth briefly mentioning the meaning of “p”.
- The formatting of table 1 should be checked.
- Row 152 (notes to table 1): “a Refers to peaks in Figure 2”, I think letter “a” refers to peaks in Figure 1.
- Table 2 (notes to Table 2): the meaning of letter “b” and “c” is not reported. I think it is the same as shown in table 1 but for clarity it is recommended to specify it.
Author Response
Cantrell and coworkers report a qualitative and semi-quantitative analysis of organosulfides extracted from steam distilled yellow onion oil through gas chromatography-mass spectrometry. Even though the extraction of organosulfides is performed with standard liquid-liquid extraction, the obtained results are noteworthy and of interest to journal readers. The paper is therefore suitable for publication in Foods after minor revision. I have however a few comments and it would be worth to take them into account.
- Most of the manuscript is written in a style that more closely resembles that of a scientific report rather than a paper. I, therefore, recommend improving the style.
- Response to reviewer: We thank the reviewer for this suggestion. We have sought the council of Assistant Editor Dana Min for guidance regarding style preference for the journal.
- Abstract section (rows 17-18): “the result was three organosulfides present…”, I suggest rephrasing this sentence in a more accurate form.
- Response to reviewer: The sentence has been changed to read “Semi-quantitative evaluation of organosulfides present in the dichloromethane extract was performed using diallyl disulfide as the internal reference standard. Three organosulfides were detected in the extract at ≥ 5 mg/kg, 18 organosulfides between 3-5 mg/kg, and six organosulfides at <3 mg/kg.”
- Row 115: it is worth briefly mentioning the meaning of “p”.
- Response to reviewer: The meaning of “p” has been described as “Poisson regression was carried out to evaluate the effect of solvent on the amount of organosulfides detected. From the regression analysis, a probability value (p-value) was calculated, where a statistically significant result correlates to a p-value ≤ 0.05.”
- The formatting of table 1 should be checked.
- Response to reviewer: We thank the reviewer for their attention to this detail. The top row of the table was shifted with respect to the other columns in the table. The formatting issue has been addressed to take into account the content in the m/z column, ensuring that values are now well aligned.
- Row 152 (notes to table 1): “a Refers to peaks in Figure 2”, I think letter “a” refers to peaks in Figure 1.
- Response to reviewer: We thank the reviewer for identifying this error. The table notes have been changed to refer to Figure 1.
- Table 2 (notes to Table 2): the meaning of letter “b” and “c” is not reported. I think it is the same as shown in table 1 but for clarity it is recommended to specify it.
- Response to reviewer: The meaning of “a”, “b”, “c”, and “d” are to indicate statistical significance among rows as stated in the Table 2 legend. The Table 2 legend has been edited to call out the values from a through d as follows, “Values across rows without a common letter (a-d) are significantly different.”
Reviewer 2 Report
In this paper, the gas chromatography-mass spectrometry (GC-MS) technique has been used to determine (qualitative and semi-quantitative analysis) organosulfides extracted from yellow onions obtained by steam distillation. Specifically, four organic solvents (dichloromethane, diethyl ether, n-pentane, hexanes), were compared for their effectiveness in the liquid-liquid extraction and quantification of organosulfides from onion oil.
The study shows interesting results and support the application of this analytical methodology for quality control measures on onion oil, however, minor revisions and some modifications need to be performed for improving the quality of the manuscript.
In my opinion, the comparison with other extraction methods (e.g. SPME, HS) it is not supported by the data provided in this work, therefore comments and/or conclusions regarding the effectiveness of these methods should be avoided in results and discussion section (only comments and comparisons between results obtained with different techniques should be discussed). Moreover, work implications should be discussed more in depth in the final part (possible practical application, limitations etc.).
Detailed comments:
Introduction
Line 36-50_This part is important to describe the current state of the art in this field, but key publications should be cited.
Line 73_ Please, delete the additional space.
Results and Discussion
Line 124_ Please, delete the additional space.
Line 127-133_ This part should be included in Material and methods part.
Line 138-140_This sentence is not clear; please rephrase or explain better.
Line 240_ I suggest changing the title of this section by deleting "versus other extraction methods".
Line 246-248_ I suggest modifying this sentence as follows: “However, quantitative analysis by headspace was not performed, nor mass spectrum fragmentation patterns for the organosulfides were obtained.”
Line 248_ Please, delete the additional space.
Line 253_ Please, delete the additional space.
Line 268-277_ Please, add some comments to discuss advantages, limitations, possible applications and future research steps.
Author Response
Line 36-50_This part is important to describe the current state of the art in this field, but key publications should be cited.
Response to reviewer: We provided a representative assessment of studies using alternative methods for organosulfide identification. It was not our intent to provide a comprehensive overview of techniques, but rather a cursory set of studies to guide the reader to seek broader knowledge if desired. To directly address the reviewer comment, we have added references [7,8] to the following sentence: “Solid phase microextraction involves the use of specialized fibers coated with extraction polymer or adsorptive particles embedded in a polymer that capture volatile molecules for analysis by gas chromatography-mass spectrometry (GC-MS).”
[7] Wardencki, W.; Michulec, M.; CuryÅ‚o, J. A Review of Theoretical and Practical Aspects of Solid-Phase Microextraction in Food Analysis. Int. J. Food Sci. Tech. 2004, 39 (703–717).
[8] D’auria, M.; Racioppi, R. HS-SPME-GC-MS Analysis of Onion (Allium Cepa L.) and Shallot (Allium Ascalonicum L.). Food Res. 2017, 1 (5), 161–165. https://doi.org/10.26656/fr.2017.5.055.
We also added reference [8] to the end of the following sentence “The extraction solvents evaluated in this study were consistent with those reported for extraction of essential oils obtained from Allium genus specimens.” Our findings were consistent with those of D’auria et al 2017.
In addition, we added reference [11] to the following sentence: “Selection of these solvents was based on literature precedent in studies of the anticancer and antibacterial properties associated with the organosulfide constituents of garlic (Allium sativum L.) and onions (Allium cepa L.) [11–14].” Reference [11] is Abe, K.; Hori, Y.; Myoda, T. Volatile Compounds of Fresh and Processed Garlic (Review). Exp. Ther. Med. 2020, 19, 1585–1593. https://doi.org/10.3892/etm.2019.8394.
Please let us know of additional “key references” that should be included in the introduction, to better describe the current state of the art in the field. We attempted to address the reviewer comment with the additional references, but recognize that we may have missed critical citations.
Line 73_ Please, delete the additional space.
Response to reviewer: The space has been deleted as requested.
Results and Discussion
Line 124_ Please, delete the additional space.
Response to reviewer: The space has been deleted as requested.
Line 127-133_ This part should be included in Material and methods part.
Response to reviewer: This change has been made as suggested by the reviewer.
Line 138-140_This sentence is not clear; please rephrase or explain better.
Response to reviewer: This sentence has been reworded as follows, “The DADS internal standard is commercially available at a purity of 78%, where the other 22% is accounted for by diallyl sulfide and diallyl trisulfide. Calibration of the DADS standard by GC-MS at five different concentrations permitted qualitative and quantitiave analysis of DADS, diallyl sulfide, and diallyl trisulfide. The diallyl sulfide and diallyl trisulfide were not utilized for the semi-quantitative assessment of organosulfides in onion oil, and were not included in Table 1.”
Line 240_ I suggest changing the title of this section by deleting "versus other extraction methods".
Response to reviewer: The title of the section has been changed as recommended.
Line 246-248_ I suggest modifying this sentence as follows: “However, quantitative analysis by headspace was not performed, nor mass spectrum fragmentation patterns for the organosulfides were obtained.”
Response to reviewer: The sentence was modified as follows: “However, quantitative analysis by headspace was not performed, nor mass spectrum fragmentation patterns for the organosulfides obtained.”
Line 248_ Please, delete the additional space.
Response to reviewer: The space has been deleted as requested.
Line 253_ Please, delete the additional space.
Response to reviewer: The space has been deleted as requested.
Line 268-277_ Please, add some comments to discuss advantages, limitations, possible applications and future research steps.
Response to reviewer: The advantages, limitations, possible applications and future research steps were included in section 3.4, which reads as follows: “The main advantage to solvent extraction of organosulfides versus either SPME or HS methods is that solvent extraction sample preparation is easy, efficient and provides semi-quantitative results. We have shown that estimates of organosulfide quantitation can be achieved with statistically significant results, as presented in Table 2. From the survey of four solvents, it was found that only seven of 29 total organosulfides identified, were detected in every solvent (compounds 4-6, 8, 10, 18, 19). Even with the superior performance of DCM to extract 27 organosulfides from onion oil, this solvent still failed to detect two of the components identified from the other three solvents, compounds 21 and 25. It is the intent of this study to provide the benefits and limitations of solvent extraction of organosulfides from onion oil to inform quality assurance measures for food flavoring products.”
Reviewer 3 Report
This manuscript titled “Determination of organosulfides from onion oil” studied the extraction efficiency of organosulfides from onion oil evaluated across four solvents (dichloromethane, diethyl ether, n-pentane, and hexanes). The method described for the extraction of organosulfides from steam distilled onion oil show common solvents to arrive at a qualitative and semi-quantitative method of analysis for agricultural products involving onions, onion oil, and secondary metabolites of Allium spp. Furthermore, the manuscript is well built and written.
However I have some comments that authors can see below:
Lines 47-48: provide a reference to support this statement
Lines 49-50: provide a reference to support this statement
Line 101: In 2.3 GC-MS Method it was not reported how volatile compounds were identified.
Lines 121-122: This sentence are part of the M&M and it has already been reported in M&M
Lines 127-133: These sentences are part of the description of the M&M
Lines 130-13: provide a reference to support this statement
improve these paragraphs: 3.1 Organosulfide Determination and Quantitation in Steam Distilled Onion Oil and 3.2 Solvent Extraction Efficiency. Many sentences are part of the description of the M&M
Line 160: add statistical analysis in figure 2
Line 182: delete “extraction of 35 mL”, was reported in M&M
Lines 185-188: provide a reference to support this statement
Lines 202-203: delete “based on triplicate extraction from 35 mL oil”, was reported in M&M
Line 215: delete “in triplicate trials”, was reported in M&M
Line 220: delete “from 35 mL onion oil”, was reported in M&M
Line 233: delete “per 35 mL oil”, was reported in M&M
Author Response
Lines 47-48: provide a reference to support this statement
Response to reviewer: Reference [9] has been associated with the statement in question. The sentence now reads as follows, “The volatile organic compound concentration can be a challenge as gas phase injection into the GC is inherently dilute [9].” Reference [9] is Falaki, F. Sample Preparation Techniques for Gas Chromatography. In Gas Chromatography - Derivatization, Sample Preparation, Application; IntechOpen, 2019; pp 1–30. https://doi.org/10.5772/intechopen.84259.
Lines 49-50: provide a reference to support this statement
Response to reviewer: The same reference applies to this sentence as the prior sentence. Reference [9] has been noted here, so the sentence now reads, “Lastly, the headspace volatilization of sample, like onion oil, requires heating, which can degrade the volatile organic compounds prior to their identification [9].”
Line 101: In 2.3 GC-MS Method it was not reported how volatile compounds were identified.
Response to reviewer: This statement was added to section 2.3, “Compounds were initially identified using the National Institute of Standards and Technologies (NIST) mass spectral library version 2.2 and confirmed where possible using literature reference.”
Lines 121-122: This sentence are part of the M&M and it has already been reported in M&M
Response to reviewer: This sentence has been removed from Section 3.1.
Lines 127-133: These sentences are part of the description of the M&M
Response to reviewer: The narrative in section 3.1 was more detailed than in the M&M. To accommodate the reviewer comment, additional content has been added to the M&M from section 3.1, and the redundant content in section 3.1 has been removed.
Lines 130-13: provide a reference to support this statement
Response to reviewer: This sentence has been moved to the M&M section and reference [20] added. The new wording is as follows, “For initial molecule identification, the National Institute of Standards and Technology (NIST) mass spectral library version 2.2 was used [20].” Reference 20 is “Shen, V.K., Siderius, D.W., Krekelberg, W.P., and Hatch, H.W., Eds., NIST Standard Reference Simulation Website, NIST Standard Reference Database Number 173, National Institute of Standards and Technology, Gaithersburg MD, 20899, http://doi.org/10.18434/T4M88Q, (retrieved 21 May 2020).”
improve these paragraphs: 3.1 Organosulfide Determination and Quantitation in Steam Distilled Onion Oil and 3.2 Solvent Extraction Efficiency. Many sentences are part of the description of the M&M
Response to reviewer: We have removed much of the redundant content to accommodate the reviewer comment.
Line 160: add statistical analysis in figure 2
Response to reviewer: To address this comment we have made the following modifications in the manuscript. The legend for Figure 2 includes a statement regarding the statistical analysis performed that reads, “The difference among solvents in (a) were analyzed by Tukey’s test with a significance of p £ 0.05.” We then added the following content to Section 3.2 Solvent Extraction Efficiency: “Statistical significance determined by Tukey’s Test indicates that DEE extraction provided the best significance when compared to the other solvents (Figure 2). The organosulfide extract obtained with DEE as the solvent was found to have significance (p £ 0.05) to the other solvents. However, no solvent was found to be significantly similar to any other solvent (i.e. hexanes with n-pentane or DCM, or n-pentane with DCM, etc.) In summary, the use of DCM afforded the extraction of a greater variety and quantity of organosulfides from the onion oil, while DEE provided the greatest statistical significance when compared to other solvents.”
The reason that the statistical analysis (letters a-d) is not added to Figure 2 is because there was not compelling statistical significance across solvents, and adding the letters to the figure detracted from the main message.
Line 182: delete “extraction of 35 mL”, was reported in M&M
Response to reviewer: Deleted as suggested by the reviewer.
Lines 185-188: provide a reference to support this statement
Response to reviewer: References [21] and [22] have been added for FEMA and GRAS. Reference 21: Oser, B. L.; Ford, R. A. GRAS Flavoring Substances 7. Recent Progress in the Consideration of Flavoring Ingredients under the Food Additives Amendment. Food Technol. 1973, 27 (11), 56–57.
Reference 22: Smith, R. L.; Cohen, S. M.; Doull, J.; Feron, V. J.; Goodman, J. I.; Marnett, L. J.; Portoghese, P. S.; Waddell, W. J.; Wagner, B. M.; Adams, T. B. GRAS Flavoring Substances 22. Flavor. Subst. 2005, 22, 24–62.
Lines 202-203: delete “based on triplicate extraction from 35 mL oil”, was reported in M&M
Response to reviewer: Deleted as suggested by the reviewer.
Line 215: delete “in triplicate trials”, was reported in M&M
Response to reviewer: Deleted as suggested by the reviewer.
Line 220: delete “from 35 mL onion oil”, was reported in M&M
Response to reviewer: Deleted as suggested by the reviewer.
Line 233: delete “per 35 mL oil”, was reported in M&M
Response to reviewer: Deleted as suggested by the reviewer.
Round 2
Reviewer 3 Report
Figure 2: Replace “Ether” with “DEE” in the figure.